# First Identification and Phylogenetic Analysis of Porcine Circovirus Type 4 in Fur Animals in Hebei, China

**DOI:** 10.3390/ani12233325

**Published:** 2022-11-28

**Authors:** Yanjin Wang, Shijie Yan, Yuting Ji, Yujie Yang, Ping Rui, Zengjun Ma, Hua-Ji Qiu, Tao Song

**Affiliations:** 1Key Laboratory of Preventive Veterinary Medicine of Hebei, College of Animal Science and Technology, Hebei Normal University of Science and Technology, Qinhuangdao 066004, China; 2State Key Laboratory of Veterinary Biotechnology, Harbin Veterinary Research Institute, Chinese Academy of Agricultural Sciences, Harbin 150069, China

**Keywords:** fur animals, porcine circovirus type 4, phylogenetic analysis

## Abstract

**Simple Summary:**

Hebei Province is an important animal husbandry region in China, especially in the fur animal farming industry. Because of poor feeding condition, fur animal farms often suffer cross-species pathogens from the surrounding pig farms. In 2019, a novel porcine circovirus, designated porcine circovirus type 4 (PCV4), was first detected in Hunan province. Therefore, a retrospective epidemiological study on PCV4-infected fur animals was performed in the frozen specimens of pigs and fur animals collected from 2015 to 2020. The results showed that PCV4 has been widely spread among pigs and fur animals in Hebei Province, China. To sum up, this is the first study to detect PCV4 in fur animals. Our findings also provide new evidence for PCV4 cross-species transmission and strengthen the hypothesis that PCV4 cross-species transmission will be a serious threat to the fur animal farming industry.

**Abstract:**

A novel circovirus called porcine circovirus type 4 (PCV4) was recently detected in pigs suffering from severe clinical diseases in Hunan province, China. There are few reports on the origin and evolution of PCV4, although some researchers have conducted epidemiological investigations of PCV4 and found that PCV4 is widespread in pigs. Based on the previous study, we detected PCV2 in farmed foxes and raccoon dogs with reproductive failure. To explore whether the PCV4 genome also exists in fur animals, we detected 137 cases admitted from fur animal farms in Hebei China between 2015 and 2020, which were characterized by inappetence, lethargy, depression, abortion, and sterility. The overall infection rate of PCV4 was 23.36% (32/137), including 20.37% (22/108) for raccoon dogs, 18.75% (3/16) for foxes, and 53.85% (7/13) for minks. Finally, five raccoon dog-origin PCV4 strains and one fox-origin PCV4 strain were sequenced in our study, whose nucleotide identities with other representative PCV4 strains varied from 96.5% to 100%. Phylogenetic analysis based on the complete genomes of PCV4 strains indicated a close relationship with those of PCV4 strains identified from pigs. To our knowledge, this is the first study to detect PCV4 in fur animals. Interestingly, we also identified PCV4 in a mixed farm (feeding pigs and raccoon dogs at the same time). In summary, our findings extend the understanding of the molecular epidemiology of PCV4 and provide new evidence for its cross-species transmission.

## 1. Introduction

Porcine circovirus (PCV) is a small, non-enveloped, single-stranded circular DNA virus, which belongs to the genus *Circovirus* within the family *Circoviridae*, with a genome of approximately 1.7 kb in length [1,2,3]. The PCV genome encodes two major open reading frames (ORFs), namely ORF1 and ORF2 [4,5]. ORF1 encodes the replicase (Rep) protein and ORF2 encodes the capsid (Cap) protein. ORF2 shows more genetic variation and is often used for genetic variation analysis. At present, PCV1, PCV2, and PCV3 have been identified in pigs [6,7,8]. PCV1 was first identified as a cell culture contaminant and was confirmed to be non-pathogenic to pigs [6]. PCV2 has been recognized as a ubiquitous pathogen that related to PCV-associated disease, including postweaning multisystemic wasting syndrome (PMWS), porcine dermatitis and nephropathy syndrome (PDNS), as well as reproductive disorders [9,10,11]. PCV2 remains one of the major pathogens that pose a threat to the global swine industry [12,13]. PCV3 was proposed as a new porcine circovirus through metagenomic sequencing in 2016, which was detected in pigs affected by cardiac pathology and PCV systemic disease [14]. It has occurred worldwide up to now and its pathogenicity remains unclear, but cloned PCV3 has been reported to potentially result in diseases similar to PDNS in pigs [15].

In 2019, a novel porcine circovirus designated porcine circovirus type 4 (PCV4) was first detected in Hunan province, followed by Henan, Shanxi and Jiangsu province [16,17,18]. Subsequently, the virus has also been identified in Guangxi, Jiangxi, Inner Mongolia and other provinces in China and South Korea [19,20,21,22]. At present, it has been identified that the length of the complete PCV4 genome sequence is 1770 bp and contains two main ORF, which encode the 296-amino-acid Rep protein and 228-amino-acid Cap protein, respectively [16,23]. A recent retrospective study reported that PCV4 could be detected in the sera collected in 2008, indicating that PCV4 has been spread in China at least 14 years [24]. At present, PCV4 has not been successfully isolated, but recent studies have indicated that rescued PCV4 was pathogenic in piglets [25]. To sum up, PCV4 has been widely spread in China and may pose a serious threat to swine industry.

PCV4 is associated with respiratory signs, enteric signs, diarrhea, and PDNS in pigs [16]. Generally, pigs are the natural reservoirs of PCV, but several reports suggested that PCV2 and PCV3 were detected in non-porcine species, and its mechanism of cross-species transmission remain to be revealed [26,27,28]. However, little is known about the cross-species transmission of PCV4. A recent study confirmed that PCV4 can infect dairy cows and the positive rate was 2.22%, which were negative for PCV2 and PCV3 [26]. Besides, genetic findings show that PCV4 shares the highest genomic identity with Mink circovirus. As we all know, China has increased to become one of the biggest fur animal farming countries, where mink is an important domestic fur animal species, in addition to fox and raccoon dog. In China, the small-scale family breeding model is the dominant breeding model of fur animals, which is often mixed with other domestic animals, such as pigs. This breeding mode has created favorable conditions for the cross-species transmission of porcine viruses to fur animals. In our previous study, we identified PCV2 in farmed foxes and raccoon dogs with reproductive failure [27,28]. Therefore, our study focused on the circumstances of PCV4 transmission between pigs and fur animals. Here, we detected PCV4 in mink, fox, and raccoon dog. This is the first study to detect PCV4 in fur animals and to report their genetic characteristics, extending our understanding of the molecular epidemiology of PCV4. This work provided new supported evidence of PCV4 cross-species transmission. Much attention should be paid to PCV4 cross-species transmission.

## 2. Materials and Methods

### 2.1. Sample Collection and Treatment

A total of 137 clinical samples from sick or dead fur animals were collected from fur animal farms in Hebei China between 2015 and 2020, which were characterized by inappetence, lethargy, depression, abortion, and sterility (Appendix A). Sample types consist of the sera, spleens, lungs, brains, livers, kidneys, lymph nodes, and small intestines. For tracing the origin of fur-animal PCV4 strains, we collected 220 serum samples of diseased pigs surrounding the fur animal farms above, which were characterized by inappetence, emaciation, and less exercise. All the tissue samples were collected aseptically and homogenized in sterilized phosphate-buffered saline (PBS), then centrifuged at 5000× *g* for 5 min. The supernatant was utilized for genome extraction immediately or stored at −80 °C for further detection.

### 2.2. Viral Genome Extract and Detection

Viral DNA was extracted from each sample using a DNA/RNA extraction kit (Luoyang Laipson, Henan, China) following the manufacturer’s instructions. All the above samples were screened for the presence of PCV4 by PCR assay (Table 1), and the positive samples were selected and amplified with whole gene primers as previously described [16]. PCV2 and PCV3 were detected by a dual nano-PCR for PCV2 and PCV3, which comprised of 95 °C for 5 min, 35 cycles of denaturation at 95 °C for 30 s, annealing at 55 °C for 30 s, and extension at 72 °C for 1 min, ended with a final extension at 72 °C for 10 min. PCR protocols were executed in a 25 μL reaction volume: 12.5 μL of 2×Master Mix (dye) (CWBio), 2 μL of template DNA, 1 μL of primer, 2 μL of nano and ddH_2_O to a total volume of 25 μL.

### 2.3. Viral Genome Sequencing

The PCV4 genome sequences were amplified using a pair of special primers, as described previously [16]. The PCR products for the whole genome sequencing were gel-purified, then cloned into pMD18-T vector and sequenced in both directions at Tianjin TSINGKE (China). The PCV4 complete sequences were assembled using the SeqMan v7.1.0 program (Lasergene, DNAStar, Madison, WI, USA).

### 2.4. Sequence Alignment and Phylogenetic Analysis

The phylogenetic analysis of the PCV4 strains obtained in this study were performed combined with the reference strains which were downloaded from GenBank (Appendix A). DNA sequences were compiled and edited using DNAStar software and multiple sequence alignment was performed using ClustalW. Sequence homology analysis was performed using the MegAlign program (Lasergene, DNAStar, Madison, WI, USA). Neighbor-joining phylogenetic trees for the complete genome were constructed using MEGA X with 1000 bootstrap replicates.

## 3. Results

### 3.1. First Detected PCV4 in Fur Animals

In this study, 137 collected samples of fur animals, including 13 minks, 16 foxes, and 108 raccoon dogs, were detected. Finally, 32 (23.36%) samples were positive for PCV4, including seven (53.85%) mink samples, three (18.75%) fox samples, and 22 (20.37%) raccoon dog samples (Table 2). Results showed that PCV4 has spread widely in Hebei Province, China, and this is the first study to report the existence of PCV4 detected in fur animals. In addition, the prevalence of PCV4 in diseased mink, with symptoms of inappetence, lethargy, depression, abortion, and sterility, is higher than that in fox and raccoon dogs. The tracing study revealed that PCV4 was first detected in samples collected in 2015, proving that PCV4 spread in Hebei Province, China since 2015 or even earlier. In addition, according to Zhang et al. [16], PCV4 was not only detected in pigs that had severe clinical signs, including respiratory disease, enteritis, and PDNS, but also in piglets with neurological symptoms, suggesting that PCV4 may have multiple tissue tropism. In our study, we also detected PCV4 in different tissue samples (Appendix A): the sera (four out of 12), spleens (eight out of 43), lungs (10 out of 50), livers (two out of nine), kidneys (six out of nine), lymph nodes (one out of seven), and small intestines (one out of four), which confirmed that PCV4 has multiple tissue tropisms. Among these tissues, the positive rate of PCV4 in kidneys, lungs and spleens higher than other tissues. So, the kidney, lung, and spleen may be the main organs attacked by PCV4 in fur animals. At last, six strains of PCV4 complete genome sequences 1770 nt in length were successfully amplified in these 32 positive samples, which contain five raccoon dog-origin and one fox-origin PCV4 strains. These fur-animal-origin PCV4 strains were named Hebei-Rac1 to Rac5 and Hebei-Fox1 respectively and were deposited in the GenBank under the accession numbers MW262979 to MW262984, respectively.

### 3.2. PCV4 Co-Infection with PCV2 and PCV3 in Fur Animals

PCV2 is one of the most important economic pig pathogens in China and all over the world. It is reported that PCV2 and PCV3 often have co-infection status and cross-species transmission. In order to understand the prevalence of PCV4 and its co-infection with other type of circovirus in fur animals, a retrospective investigation was designed to detect PCV2, PCV3, and PCV4 in diseased fur animals of Hebei Province. For the co-infection assay, the positive rates of PCV4 co-infection with PCV2 and PCV3 was 9.49% (13/137) and 10.22% (14/137), respectively (Table 3). There was no significant co-infection status between PCV4 and other PCVs (PCV2 and PCV3), which proved that PCV4 might propagate independently as previously suspected. Moreover, the prevalence of co-infection between PCV4 and other PCVs (PCV2 and PCV3) in mink is also higher than that of other two fur animals.

### 3.3. Tracking Study of PCV4 in Nearby Pig Farms

For tracing the origin of fur-animal PCV4 strains, a retrospective epidemiological study was conducted in the surrounding pig farms on a large scale. We selected 220 serum samples of pigs, which were submitted to our laboratory for detection. This revealed that the PCV4 positive rate of pigs was 20.45% (45 out of 220). Strongly positive samples of pigs were selected for sequencing the PCV4 complete genome sequence. Finally, six pig-origin PCV4 strains were sequenced and designated as PCV4 Hebei1–6 respectively, which were deposited into the GenBank database under accession numbers MW262973–MW262978, respectively.

### 3.4. Sequence Alignment Analysis of PCV4 Strains

To further understand the genetic characteristics of PCV4 in Hebei Province, five raccoon dog-origin, one fox-origin, six pig-origin PCV4 and 36 representative PCV4 strains were used for genome sequence analysis. Pairwise-sequence comparisons of the complete genomes showed that the genome nucleotide similarity between twelve PCV4 strains with other PCV4 representative strains varied from 96.5 to 100% (Figure 1). Within these twelve PCV4 strains, fur animal-origin PCV4 strain shared 99.7 to 99.9% nucleotide identities with pig-origin PCV4 strain detected around the fur animal farm (Figure 1). The high nucleotide homology among currently identified PCV4 strains showed that PCV4 had little variation. Notably, Hebei-Rac5 and Hebei-5 are identified from the same farm, where both raccoon dogs and pigs are raised at the same time. The homology among Hebei-Rac5 and Hebei-5 is 99.8%, but this is not the highest. This shows that PCV4 has multiple epidemiological characteristics in Hebei Province.

### 3.5. Phylogenetic Analysis of PCV4 Strains

Genome-based phylogenetic tree analysis showed that six fur animal-origin, six pig-origin PCV4 strains, and 86 PCVs reference strains formed four distinct clusters and all the obtained strains in this study belonged to PCV4 (Figure 2). All these obtained PCV4 strains were relatively distant from PCV1, PCV2, and PCV3, but shared a higher sequence similarity with Mink circovirus MiCV-DL13 and SD16. Furthermore, to clarify the genetic subtype of twelve PCV4 strains in this study, another phylogenetic tree was produced with 36 representative PCV4 strains (Figure 3). The results showed all these 12 PCV4 strains belonged to subtype PCV4b with Strain Henan-LY1-2019. To test for possible recombination, SimPlot software was used to perform similarity comparisons and bootscanning analysis. The results showed there were no recombination regions or sites among these fur animal-origin PCV4 strains.

## 4. Discussion

Hebei Province, located in the north of China, covers an area of 188,800 km^2^ and is an important animal husbandry region in China. PCV2 has been among the most important porcine viruses in China [29,30]. Recent studies have suggested that PCV2 and PCV3 were detected in non-porcine species. Our laboratory focused on the cross-species transmission of circoviruses, where PCV2 had been identified by our laboratory in foxes and raccoon dogs with aborted or sterile signs [27,28]. This indicated that PCV2 is a potential threat to fur animal breeding. Regarding the retrospective prevalence of PCV2 in fur animals, we speculated that the infection of PCV2 in fur animals was due to the current breeding mode of fur animals. In China, the small-scale family breeding model was the dominant breeding model of fur animals, as reflected in the close proximity of pig and fur animal farms or mixed breeding. This breeding mode promoted the cross-species transmission of circoviruses.

In 2019, a new type of circovirus named PCV4 was identified in pigs with severe clinical symptoms and genetic findings, which showed high genomic identity to Mink circovirus. Therefore, a retrospective investigation was designed on the prevalence of PCV4 in fur animals. The prevalence of PCV4 (23.36%) determined in the present study was lower than that detected in Henan and Shanxi Provinces (25.4%), and significantly higher than that detected in Inner Mongolia (1.6%), Guangxi Province (5.1%), and Hunan Province (12.8%), China. Results showed that PCV4 had spread widely in Hebei Province, China. Notably, the earliest PCV4 positive samples came from 2015, indicating that PCV4 existed in fur animals of Hebei Province, China for at least seven years. To our knowledge, this is the first study to identify PCV4 in fur animals. In addition, we evaluated the co-infection of PCV4 with other PCVs.

Interestingly, there is no significant co-infection status between PCV4, PCV2, and PCV3, which indicated that PCV4 may act as an individual pathogen related to disease. This is important for understanding the genetic diversity and cross-species transmission of the current PCV4. The phylogenetic analysis demonstrated PCV4 genome was more similar to Mink circovirus genome than that of PCV1-3, indicating that the occurrence of PCV4 may have evolved from Mink circovirus. Co-infection events of different PCVs increased the possibility of novel PCV occurrence. Conversely, genetic evolution was more conducive to PCVs co-infection.

For tracing the origin of these raccoon dog-origin and fox-origin PCV4 strains, we detected 220 sera samples of pig farms near fur animal farms, which submitted in our laboratory. As expected, we detected PCV4 prevalence (20.45%) in sera samples from nearby pig farms and obtained six complete genomes of PCV4 strains. The homology alignment revealed that the genome sequences of pig-origin PCV4 strains shared 99.7 to 99.9% nucleotide identities with those of six fur animal-origin PCV4 strains, which indicated the possibility cross-species transmission of PCV4. Horizontal transmission was confirmed to be the most efficient route of PCV4 spreading between infectious and susceptible individuals, but the initial host of PCV4 as well as the transmission direction remain unknowable. Notwithstanding, the current evidence does not indicate that the prevalent PCV4 in fur animals originated from pigs. Rather, PCV4 may have originated from minks.

## 5. Conclusions

In conclusion, this study provided the first evidence of PCV4 prevalence in fur animals in northern China. The genetic analysis and epidemiological investigation of these fur animals-originated PCV4 stains are helpful for elucidating the genetic evolution and epidemiology of fur animal-origin PCV4. It also provides the basis for the prevention and control of PCV4 in the future. However, we can be sure that traditional mixed feeding model of fur animals with other species represents a severe threat to the fur animal breeding industry, and vigorous measures should be taken as soon as possible to prevent PCV4 cross-species transmission.

## Figures and Tables

**Figure 1 animals-12-03325-f001:**
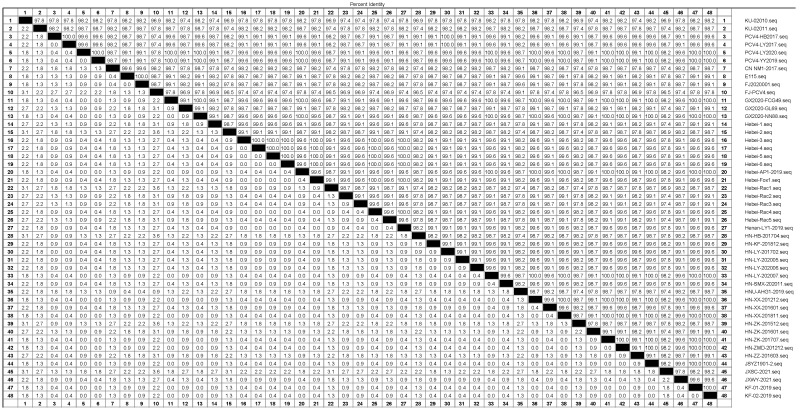
Sequence distance of the complete genome sequences between twelve PCV4 strains obtained in this study and representative PCV4 strains.

**Figure 2 animals-12-03325-f002:**
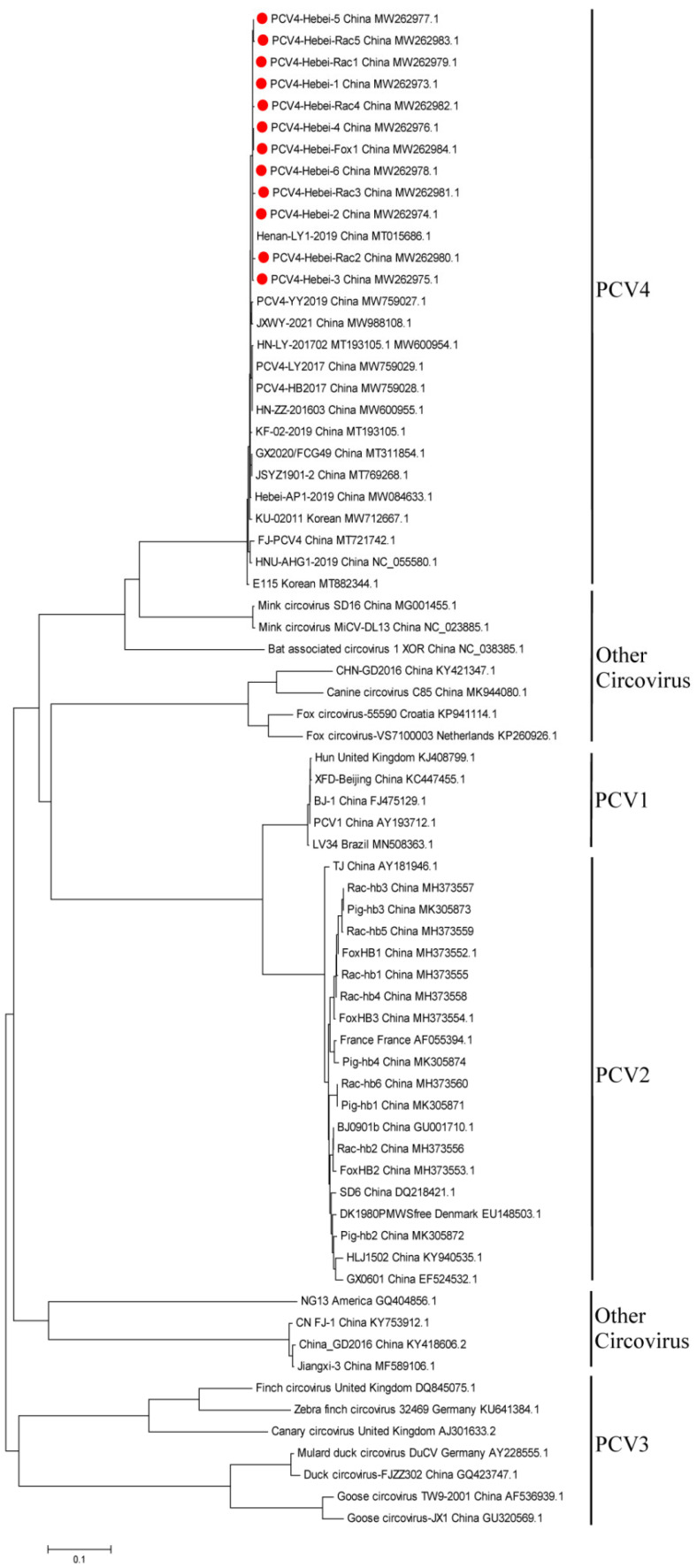
Phylogenetic tree of the whole genome sequences of PCVs. The tree was constructed using MEGA X software with neighbor-joining methods and 1000 replicate sets on bootstrap analysis. The twelve PCV4 strains obtained in this study were labelled with red circular.

**Figure 3 animals-12-03325-f003:**
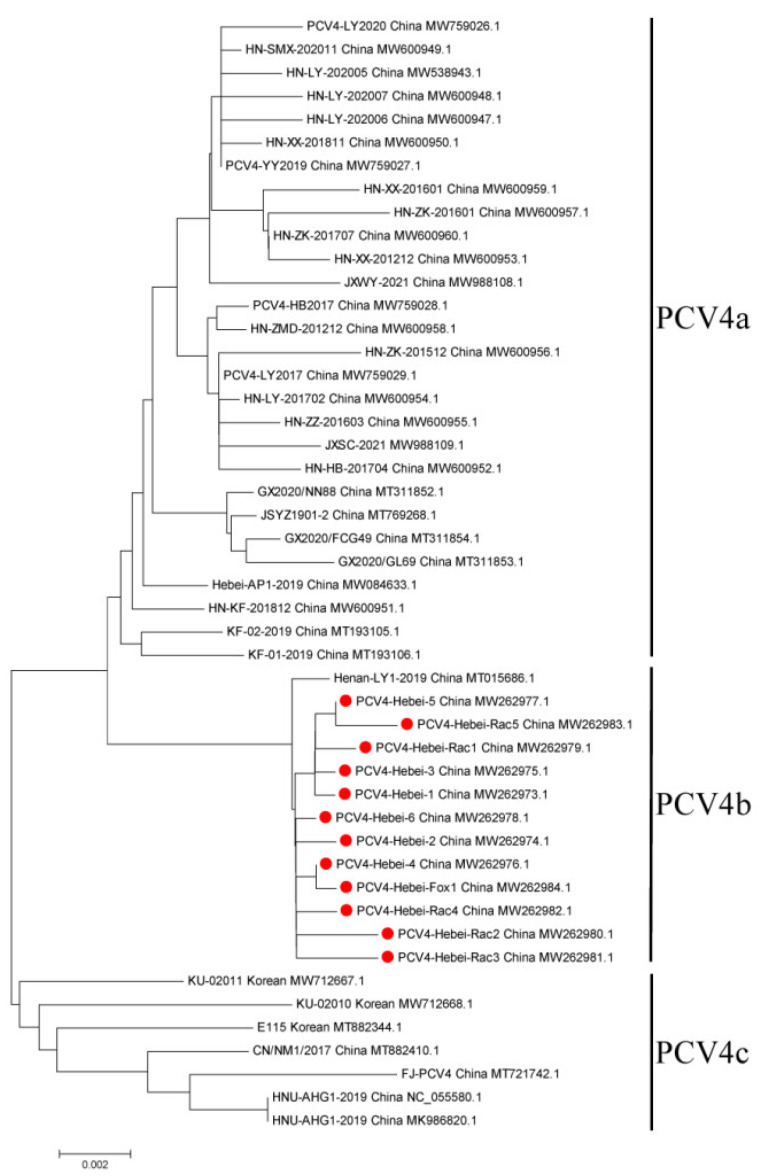
Phylogenetic trees of the whole genome sequences of 48 PCV4 strains including 12 strains obtained in this study and 36 other HEV representative strains. The tree was constructed using MEGA X software with neighbor-joining methods and 1000 replicate sets on bootstrap analysis. The twelve PCV4 strains obtained in this study were labelled with red circular.

**Table 1 animals-12-03325-t001:** Primers used in this study.

Primer Name	Sequence	Purpose	Size (bp)
PCV2-F	CTCCGGTAAGCGCCTCCTTG	Detection	883
PCV2-R	GATAGAGAGCTTCTACAGCTG		
PCV3-F	TTACTTAGAGAACGGACTTGTAACG	Detection	649
PCV3-R	AAATAGACACAGAGCTATATTCAG		
PCV4-F	CGGTGAGTTCCCGTCTGTATTT	Detection	391
PCV4-R	TCACGGGCCACTTCACTCAT		

**Table 2 animals-12-03325-t002:** PCR identification of PCV4 infection in diseased fur animals in Hebei Province of China.

Fur Animal Species	Total Samples	PCV4-Positive Samples	PCV4-Positive Rate
Raccoon dog	108	22	20.37%
Fox	16	3	18.75%
Mink	13	7	53.85%
Total	137	32	23.36%

**Table 3 animals-12-03325-t003:** PCV4 co-infection with PCV2 and PCV3 in diseased fur animals.

Fur Animal Species	Total Samples	Samples Co-Infectied with PCV4			
		PCV2		PCV3	
Raccoon dog	108	8/108	7.41%	10/108	9.26%
Fox	16	1/16	6.25%	1/16	6.25%
Mink	13	4/13	30.77%	3/13	23.08%
Total	137	13/137	9.49%	14/137	10.22%

## Data Availability

Not applicable.

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
