# Peer review of "First Identification and Phylogenetic Analysis of Porcine Circovirus Type 4 in Fur Animals in Hebei, China"

_animals, 2022, doi:10.3390/ani12233325_

Round 1

Reviewer 1 Report

The manuscript provided first evidence of the porcine circovirus 4(PCV4), an emerging pathogen that has raised a concern in swine industry, prevalence in fur animals in northern China, and the possibly cross-species transmission between fur animals and swine. The manuscript is written well and deals with the current topic. I recommend it for publication in Animals.

Minor Comments:

1. Need more detail clinical status of these PCV-4 infected fur animals. Better to understanding the role of PCV4 in disease. And have any clinical status of the swine infected with PCV-4 in the swine and racoon dog mixed raising farm?

2.Please add the results of the recombinant analysis, Because these new sequences of PCV4 comes from a new species.

3.Line 106: ddH2O? 2 should be lowercase.

4. genetic variation observed was not discussed at all. Whether the cap gene is mutated in fur animals when PCV4 is transmitted across species.

5. Need more discussion: a relationship between PCV4 genetics and co-infection.

6. In the sentence in 172-173 Pairwise-sequence comparisons of the complete genomes showed that the genome nucleotide similarity of twelve PCV4 strains obtained in this study with PCV4 representative isolates ranged from 37.6%~99.9%. ” It should be a serious wrong for the lowest genome nucleotide similarity only 37.6%! And author provided with no data both in manuscript and supplementary at all.

Author Response

Reviewer 1

The manuscript provided first evidence of the porcine circovirus 4(PCV4), an emerging pathogen that has raised a concern in swine industry, prevalence in fur animals in northern China, and the possibly cross-species transmission between fur animals and swine. The manuscript is written well and deals with the current topic. I recommend it for publication in Animals.

Reply: We greatly appreciate the valuable comments and constructive suggestions of Reviewer #1. We have revised the manuscript as suggested.

Minor Comments:

  1. Need more detail clinical status of these PCV-4 infected fur animals. Better to understanding the role of PCV4 in disease. And have any clinical status of the swine infected with PCV-4 in the swine and racoon dog mixed raising farm?

Reply: We added more detail clinical status of these PCV4 infected fur animals and swine in the manuscript as suggested (Line 94-99).

2.Please add the results of the recombinant analysis, Because these new sequences of PCV4 comes from a new species.

Reply: Compared with PCV1-3, PCV4 is a novel virus of the genus Circovirus. The complete genome sequences of twelve PCV4 strains obtained in this study showed high homology of 99.7%~99.9% with fourteen representative PCV4 strains, which has been reported (Figure 1). We have used Simplot software to product the recombination analysis, and the results shows that these twelve PCV4 strains have no homologous recombination. The recombination analysis description have been added to the manuscript (Line 194~197).

3.Line 106: ddH2O? 2 should be lowercase.

Reply: Thank the reviewer for pointing out this omission, we have revised it in the manuscript.

  1. genetic variation observed was not discussed at all. Whether the cap gene is mutated in fur animals when PCV4 is transmitted across species.

Reply: Thanks very much for this professional and scientific suggestion. We have analyzed the twelve PCV4 strains obtained in this study using Simplot software, and the results showed that the twelve PCV4 strains have no homologous recombination among the 35 representative PCV4 strains, and the modifications were made in the manuscript (Line 194~197).

  1. Need more discussion: a relationship between PCV4 genetics and co-infection.

Reply: We added more discussion on the relationship between PCV4 genetics and co-infection in the manuscript as suggested (Line 225~229).

  1. In the sentence in 172-173 “Pairwise-sequence comparisons of the complete genomes showed that the genome nucleotide similarity of twelve PCV4 strains obtained in this study with PCV4 representative isolates ranged from 37.6%~99.9%. ” It should be a serious wrong for the lowest genome nucleotide similarity only 37.6%! And author provided with no data both in manuscript and supplementary at all.

Reply: We feel very guilty about this mistake. We have analyzed the nucleotide similarity of PCV4 with that of PCV1-3 and other circoviruses, which showed the lowest homology was only 37.6%. Because it is little revelent to this paper, we have removed the figure of the sequence alignment analysis results of PCV4 and other circoviruses. But forget to resive the manuscript. On the reviewer’s suggestion, We have modified this result in the manuscript.

Reviewer 2 Report

The manuscript entitled "First Identification and Phylogenetic Analysis of Porcine Circovirus Type 4 in Fur Animals in Hebei, China" (Manuscript ID animals-2031068) described the first identification and phylogenetic analysis of PCV4 in fur animals including raccoon dogs, foxes and minks. The results of this study enriched the epidemiological information of PCV4. My specific comments on the manuscript are as follows:

1. Ethical approval and the approval number approved by the Animal Ethics Committee should be provided to prove the legitimacy of sampling during the collection of animal samples.

2. The information for the 137 clinical samples is incomplete, the coordinates of the sampling site should be provided. Whether the same fur animals come from the same farm. What are the main clinical symptoms of the fur animals that were sampled.

3. The authors did not write clearly the primer sequences used for amplification of the PCV4 whole genome sequence. The primer pair listed in Table 1 is just a detection primer.

4. The authors did not provide any evidence to show their primers to amplify the whole genome of PCV4 or detect PCV4 were highly conserved, therefore, I cannot judge whether the primer used by the author can reliably detect the majority of PCV4 strains.

5. During phylogenetic analysis of PCV4 strains, the authors said the genome nucleotide similarity of the twelve PCV4 strains obtained in the study with PCV4 representative isolates ranged from 37.6%~99.9%. Those sequences with low homology are probably not PCV4. Please explain this.

6. The name of the figure legends of Figure 2 is inaccurate, because in addition to PCV4, PCV of other genotypes (PCV1, PCV2 and PCV3) were also included in the figure. Moreover, please label the two figures in Figure 2.In addition, please indicate the national source of each strain in the figure.

7. The language of the whole manuscript needs to be edited by an English-speaking professional.

Author Response

Reviewer 2

The manuscript entitled "First Identification and Phylogenetic Analysis of Porcine Circovirus Type 4 in Fur Animals in Hebei, China" (Manuscript ID animals-2031068) described the first identification and phylogenetic analysis of PCV4 in fur animals including raccoon dogs, foxes and minks. The results of this study enriched the epidemiological information of PCV4. My specific comments on the manuscript are as follows:

  1. Ethical approval and the approval number approved by the Animal Ethics Committee should be provided to prove the legitimacy of sampling during the collection of animal samples.

Reply: Thank the reviewer for this professional and scientific suggestion. In our research, all the specimens were collected from fur animals farms and pig farms in Qinhuangdao of China between 2015 and 2020. These specimens were sent to Animal Hospital of Hebei Normal University of Science and Technology by farm owners for testing. We were allowed to conduct the the retrospective analysis by all the farm owners, and have provided the informed consents of animal owners, which had been submitted to the website of Animals. As a licensed veterinary physician, ethical approval was granted by the Administration Bureau of Animal Husbandry and Veterinary Medicine of Hebei.

  1. The information for the 137 clinical samples is incomplete, the coordinates of the sampling site should be provided. Whether the same fur animals come from the same farm. What are the main clinical symptoms of the fur animals that were sampled.

Reply: According to the reviewer’s suggestion, we have added the sample information in our manuscript (Line 94~99) and Table S1.

  1. The authors did not write clearly the primer sequences used for amplification of the PCV4 whole genome sequence. The primer pair listed in Table 1 is just a detection primer.

Reply: We have amplified the complete sequences of PCV4 by referring to the primer sequence in the published articles. In the manuscript, we have introduced and cited this article. So, we are not listed in the Table 1.

  1. The authors did not provide any evidence to show their primers to amplify the whole genome of PCV4 or detect PCV4 were highly conserved, therefore, I cannot judge whether the primer used by the author can reliably detect the majority of PCV4 strains.

Reply: The primers to amplify the whole genome of PCV4 or detect PCV4 were highly conserved, which has been reported [16]. At the same time, from our genome homology results of reported PCV4, it can be concluded that PCV4 is highly conservative. So, the primers used in this study can reliably detect and amplify the whole genome of the majority of PCV4 strains.

[16] Zhang, H. H., Hu, W. Q., Li, J. Y., Liu, T. N., Zhou, J. Y., Opriessnig, T., & Xiao, C. T. (2020). Novel circovirus species identified in farmed pigs designated as Porcine circovirus 4, Hunan province, China. Transboundary and emerging diseases, 67(3), 1057–1061.

  1. During phylogenetic analysis of PCV4 strains, the authors said the genome nucleotide similarity of the twelve PCV4 strains obtained in the study with PCV4 representative isolates ranged from 37.6%~99.9%. Those sequences with low homology are probably not PCV4. Please explain this.

Reply: We feel very guilty about this mistake. We have analyzed the nucleotide similarity of PCV4 with that of PCV1-3 and other circoviruses, which showed the lowest homology was only 37.6%. Because it is little revelent to this paper, we have removed the figure of the sequence alignment analysis results of PCV4 and other circoviruses. But forget to resive the manuscript. On the reviewer’s suggestion, We have modified this result in the manuscript.

  1. The name of the figure legends of Figure 2 is inaccurate, because in addition to PCV4, PCV of other genotypes (PCV1, PCV2 and PCV3) were also included in the figure. Moreover, please label the two figures in Figure 2.In addition, please indicate the national source of each strain in the figure.

Reply: Thanks very much for this professional and scientific suggestion. On the reviewer’s suggestion, we have revised the manuscript (Line 192~200, Figure 2 and 3).

  1. The language of the whole manuscript needs to be edited by an English-speaking professional.

Reply: Thank the reviewer for this professional and scientific suggestion. According to the reviewer’s suggestion, our manuscript has been revised by a native English speaker.

Reviewer 3 Report

A very interesting paper describing an inter-species potential of infection with PCV4. The presented data showed that PCV4 which is a naive virus of pigs may infect raccoon dogs, minks and foxes. The presented data are reliable and extremely novel. The paper is well written. In view of my expertise it may be accepted in the current form. 

Author Response

Reviewer 3

A very interesting paper describing an inter-species potential of infection with PCV4. The presented data showed that PCV4 which is a naive virus of pigs may infect raccoon dogs, minks and foxes. The presented data are reliable and extremely novel. The paper is well written. In view of my expertise it may be accepted in the current form.

Reply: Thank you very much for reviewing our manuscript. We have also checked the full text of the manuscript and made corrections accordingly.